# Visualization and molecular characterization of whole-brain vascular networks with capillary resolution

Takeyuki Miyawaki [1✉], Shota Morikawa[1], Etsuo A. Susaki[2,3], Ai Nakashima[1], Haruki Takeuchi[1,4], Shun Yamaguchi[5,6], Hiroki R. Ueda[2,3,7] & Yuji Ikegaya [1,8]

Structural elucidation and molecular scrutiny of cerebral vasculature is crucial for understanding the functions and diseases of the brain. Here, we introduce SeeNet, a method for near-complete three-dimensional visualization of cerebral vascular networks with high signal-to-noise ratios compatible with molecular phenotyping. SeeNet employs perfusion of a multifunctional crosslinker, vascular casting by temperature-controlled polymerization of hybrid hydrogels, and a bile salt-based tissue-clearing technique optimized for observation of vascular connectivity. SeeNet is capable of whole-brain visualization of molecularly characterized cerebral vasculatures at the single-microvessel level. Moreover, SeeNet reveals a hitherto unidentified vascular pathway bridging cerebral and hippocampal vessels, thus serving as a potential tool to evaluate the connectivity of cerebral vasculature.

[1] Graduate School of Pharmaceutical Sciences, The University of Tokyo, Tokyo, Japan. [2] Department of Systems Pharmacology, Graduate School of Medicine, The University of Tokyo, Tokyo, Japan. [3] Laboratory for Synthetic Biology, RIKEN Center for Biosystems Dynamics Research, Osaka, Japan. [4] Social Cooperation Program of Evolutional Chemical Safety Assessment System, LECSAS, Graduate School of Pharmaceutical Sciences, The University of Tokyo, Tokyo, Japan. [5] Department of Morphological Neuroscience, Graduate School of Medicine, Gifu University, Gifu, Japan. [6] Center for Highly Advanced Integration of Nano and Life Sciences, Gifu University, Gifu, Japan. [7] International Research Center for Neurointelligence (WPI-IRCN), UTIAS, The University of Tokyo, Tokyo, Japan. [8] Center for Information and Neural Networks, National Institute of Information and Communications Technology, Osaka, Japan. ✉email: takeyuki.miyawaki@gmail.com

Cerebral vasculature is an anatomically and histologically inhomogeneous three-dimensional (3D) traffic network for blood cells, nutrients, oxygen, metabolic wastes, signaling molecules, etc.[1–3]. Although structural and molecular analyses of cerebral vasculature are promising for providing fundamental insights into cerebral circulation and cerebrovascular diseases, which are major causes of death and disability[4], few studies have captured detailed structural information of molecularly identified vascular networks due to technical limitations (Supplementary Tables 1 and 2)[5–21].

Here, we introduce SeeNet, a method for 3D visualization and molecular characterization of the entire vascular network, including virtually all microvessels, in the whole brain. SeeNet includes (i) intravascular perfusion of a blood–brain barrier (BBB)-impermeable fluorescent crosslinker, (ii) thermally controlled polymerization of a hybrid hydrogel, and (iii) bile salt-based tissue clearing.

## Results

### Thermally controlled vascular casting with hybrid hydrogel.

Classic methods for vascular casting using solidifying polymers, such as resin[11], fail to allow visualization of all vessels because the polymers often become clogged in microvessels. To suppress clogging, we conceived a stratagem in which polymerization was initiated after complete perfusion of cerebral vessels with monomer acrylamide and a BBB-impermeable fluorescent crosslinker. We designed a fluorescent compound termed RITC–Dex–GMA consisting of rhodamine isothiocyanate (RITC), a fluorescent group; glycidyl methacrylate (GMA), a crosslinkable group; and dextran (Dex), a macromolecule that makes the compound have a molecular weight large enough to prevent it from crossing the BBB and small enough to not clog microvessels (Fig. 1a). RITC–Dex–GMA can be synthetized through a few simple chemical reactions and can easily be purified with ethanol precipitation (Supplementary Fig. 1a, b)[22,23]. To regulate the timing of the polymerization of RITC–Dex–GMA and acrylamide, we adopted the thermal initiator VA-044; note that VA-044 triggers polymerization when heated to 37 °C (Supplementary Fig. 1c, d). After intravascular perfusion with phosphate-buffered saline (PBS) and paraformaldehyde (PFA), mice were further perfused with a mixture of RITC–Dex–GMA, acrylamide, and VA-044 at 4 °C. When the sample was maintained at 37 °C for 3 h, a hybrid hydrogel was formed; fluorescent hydrogels formed inside the vascular lumen, whereas nonfluorescent, protein-crosslinked hydrogels formed outside the lumen (Fig. 1b).

We compared the visualization performance of this method (Fig. 1c) to that of intravenous injection or transcardial perfusion of Texas Red-conjugated lectin, which are commonly used techniques for vascular staining (Supplementary Figs. 2a and 3a). The brain was thin-sectioned and immunolabeled against CD31, a marker for endothelial cells (Supplementary Figs. 2b and 3b). The signal-to-noise (SN) ratio of the vasculature casted by RITC–Dex–GMA was significantly higher than that of the vasculature stained by intravenous injection or transcardial perfusion of Texas Red-conjugated lectin (Fig. 1d, $n = 9$, 10, and 6 slices from 9, 10, and 6 mice, respectively, Dex–GMA versus lectin (i.v.): $P = 1.59 \times 10^{-10}$, $Q_{2,22} = 16.7$; Dex–GMA versus lectin (perfused): $P = 2.01 \times 10^{-5}$, $Q_{2,22} = 8.27$; $P = 2.94 \times 10^{-10}$, $F_{2,22} = 69.9$, Tukey's test after one-way ANOVA). The SN ratio of transcardially perfused lectin was significantly higher than that of intravenously injected lectin (Fig. 1d, $n = 6$ and 10 slices from 6 and 10 mice, respectively, $P = 4.53 \times 10^{-4}$, $Q_{2,22} = 6.42$, Tukey's test after one-way ANOVA). However, as previously

reported in the liver[15], vasculature staining with transcardial perfusion of lectin was spotty (Supplementary Fig. 3). The overlap ratio of RITC–Dex–GMA-casted vessels and CD31-immunopositive vessels was 98.2 ± 0.8%, which was significantly higher than the overlap ratio 59.4 ± 9.45% (intravenous injection) and 64.4 ± 7.32% (perfused) for Texas Red-conjugated lectin (Fig. 1e, mean ± SD of 9, 10, and 6 slices from 9, 10, and 6 mice, respectively, Dex–GMA versus lectin (i.v.): $P = 2.39 \times 10^{-10}$, $Q_{2,22} = 16.4$; Dex–GMA versus lectin (perfused): $P = 3.51 \times 10^{-8}$, $Q_{2,22} = 12.4$; lectin (i.v.) versus lectin (perfused): $P = 0.401$, $Q_{2,22} = 1.86$, $P = 1.59 \times 10^{-10}$, $F_{2,22} = 74.5$, Tukey's test after one-way ANOVA). Therefore, thermally controlled post hoc polymerization of perfused RITC–Dex–GMA could be used to visualize nearly all of the cerebral vasculature with a high SN ratio.

We also compared the staining performance between our method and a gelatin–fluorescein isothiocyanate (FITC)–albumin gel perfusion method[10], which is reported to be able to label cerebral vasculature with a high SN ratio (Supplementary Fig. 4a). Unexpectedly, however, we noticed that the gelatin–FITC–albumin gel perfused samples were only partially stained with anti-CD31 (Supplementary Fig. 4b–d). The percentage of CD31-positive areas in slices was 4.37 ± 0.94%, in contrast to the expected 9.25 ± 1.12% when CD31 is successfully stained (Supplementary Fig. 4e, mean ± SD of 5 and 9 slices from 5 and 9 mice, respectively, $P = 5.83 \times 10^{-6}$, $t_{12} = 7.66$, Student's t-test). A possible explanation for this incomplete staining could be that the warm PFA used in this method altered the efficiency of fixation or the antigenicity of CD31. To examine whether the antigenicity was affected, we compared the staining patterns of other protein markers for neurons (NeuN), astrocytes (GFAP), and microglia (Iba-1) in samples that were fixed using a standard method (PFA), our method (Dex–GMA), or the gelatin–FITC–albumin method (Supplementary Fig. 5a–c). The immunostaining patterns in Dex–GMA-treated samples were similar to those of PFA-fixed samples. However, in gelatin–FITC–albumin-treated samples, the immunosignals were deteriorated, and the staining patterns were not spatially even. The percentages of immunopositive areas to entire areas were similar between PFA-treated and Dex–GMA-treated samples, whereas the percentages were lower in gelatin albumin-treated samples (Supplementary Fig. 5d; $P = 3.47 \times 10^{-3}$, $F_{2,12} = 9.44$, one-way ANOVA; PFA versus Dex–GMA: $P = 0.46$, $Q_{2,12} = 1.73$; Dex–GMA versus gelatin–albumin: ***$P = 3.13 \times 10^{-3}$, $Q_{2,12} = 5.97$; PFA versus gelatin–albumin: *$P = 2.79 \times 10^{-2}$, $Q_{2,12} = 4.238$, Tukey's test. Supplementary Fig. 5e; $P = 9.19 \times 10^{-3}$, $H_{2,12} = 9.38$, Kruskal–Wallis test; PFA versus Dex–GMA: $P = 0.10$, $Q_{2,12} = 0.99$; Dex–GMA versus gelatin–albumin: *$P = 2.45 \times 10^{-2}$, $Q_{2,12} = 2.61$; PFA versus gelatin–albumin: *$P = 2.45 \times 10^{-2}$, $Q_{2,12} = 2.61$, Steel-Dwass test. Supplementary Fig. 5f; $P = 7.60 \times 10^{-4}$, $F_{2,12} = 13.9$, one-way ANOVA; PFA versus Dex-GMA: $P = 8.48 \times 10^{-2}$, $Q_{2,12} = 3.342$; Dex–GMA versus gelatin–albumin: ***$P = 5.48 \times 10^{-4}$, $Q_{2,12} = 7.43$; PFA versus gelatin–albumin: **$P = 3.35 \times 10^{-2}$, $Q_{2,12} = 4.09$, Tukey's test.). These data indicate that the antigenicity was preserved in Dex–GMA-treated and PFA-fixed samples but was altered in gelatin–FITC–albumin-treated samples. In the gelatin–FITC–albumin gel perfusion method, we occasionally observed clusters of CD31-positive microvessels that were not stained by FITC, suggesting that gelatin–FITC–albumin gels were clogged in some brain regions (Supplementary Fig. 4f). To quantitatively compare the staining performance between the two methods, we labeled endothelial cells using the intravenously injected lectin-conjugated dye for a reference (Supplementary Fig. 6a, b). Unlike anti-CD31 staining, the performances of lectin labeling were similar between the two methods, possibly because the labeling was carried out before the perfusion of warmed PFA and the change in the antigenicity

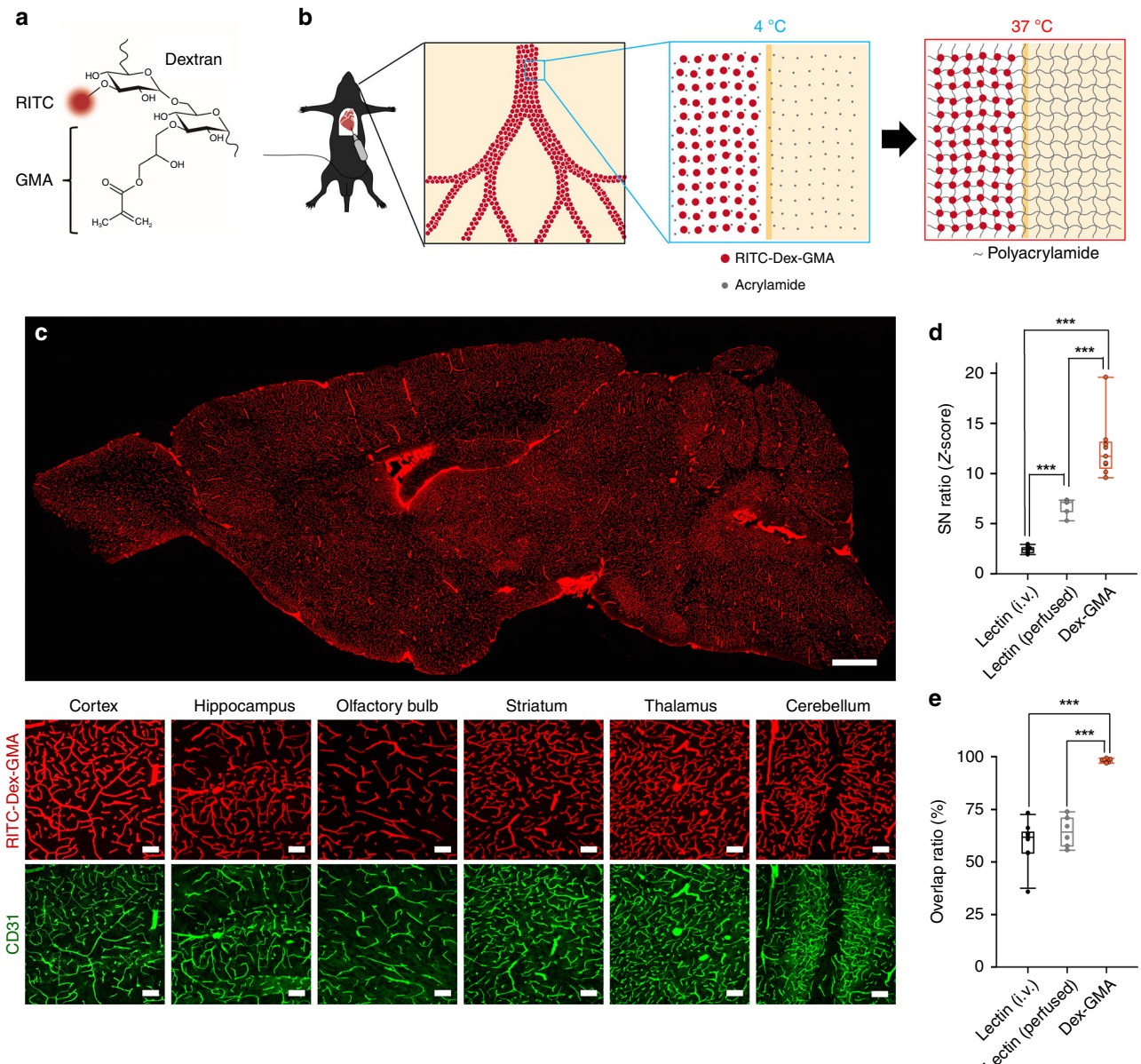

**Fig. 1 Near-complete visualization of cerebral vasculature using thermally controlled polymerization of a BBB-impermeable fluorescent crosslinker. a** Design of the BBB impermeable fluorescent crosslinker RITC–Dex–GMA. The fluorophore RITC and the crosslinkable group GMA are chemically linked through dextran, a BBB-impermeable molecule. **b** Strategic concept of vascular casting using a hybrid hydrogel. Red circles and gray dots indicate RITC–Dex–GMA and acrylamide, respectively. **c** Top: A confocal image of a sagittal section of an adult mouse brain treated with vascular casting using RITC–Dex–GMA, acrylamide, and VA-044. Scale bar = 1 mm. Bottom: Magnified images of RITC–Dex–GMA-casted vasculature of different brain regions (red) and endothelial cells counterstained using anti-CD31 in the same fields (green). Scale bar = 100 μm. **d** SN ratios of the fluorescence intensities in lectin-based and Dex–GMA-based visualization ($n = 10$, 6, and 9 slices from 10, 6, and 9 mice, respectively; box plots indicate median and 25–75% interquartile ranges; whiskers cover 10−90% quantiles. $P = 2.94 \times 10^{-10}$, $F_{2,22} = 69.9$, one-way ANOVA; Dex-GMA versus lectin (i.v.); ***$P = 1.59 \times 10^{-10}$, $Q_{2,22} = 16.7$; Dex-GMA versus lectin (perfused); ***$P = 2.01 \times 10^{-5}$, $Q_{2,22} = 8.27$; lectin (i.v.) versus lectin (perfused): ***$P = 4.53 \times 10^{-4}$, $Q_{2,22} = 6.42$, Tukey's test). **e** Overlap ratios between anti-CD31 immunoreactivity and Texas Red/RITC fluorescence in lectin-based and Dex–GMA-based visualization ($59.4 \pm 9.45\%$, $64.4 \pm 7.32\%$, and $98.2 \pm 0.8\%$, mean ± SD of 10, 6, and 9 slices from 10, 6, and 9 mice, respectively; box plots indicate median and 25−75% interquartile ranges; whiskers cover 10−90% quantiles. $P = 1.59 \times 10^{-10}$, $F_{2,22} = 74.5$, one-way ANOVA; Dex-GMA versus lectin (i.v.); ***$P = 2.39 \times 10^{-10}$, $Q_{2,22} = 16.4$; Dex-GMA versus lectin (perfused); ***$P = 3.51 \times 10^{-8}$, $Q_{2,22} = 12.4$; lectin (i.v.) versus lectin (perfused): $P = 0.401$, $Q_{2,22} = 1.86$, Tukey's test). Source data are provided as a Source Data file.

(Supplementary Fig. 6c, $10.9 \pm 0.41\%$ vs. $10.9 \pm 0.54\%$, mean ± SD of five slices from five mice each, $P = 0.892$, $t_8 = 0.14$, Student's $t$-test). Again, the clusters of lectin-positive vessels that were not stained by FITC were occasionally observed in gelatin–FITC-treated samples (Supplementary Fig. 6a), and the

performance of vasculature staining was higher in the FITC–Dex–GMA-based protocol (Supplementary Fig. 6d, $95.9 \pm 1.97\%$ vs. $98.3 \pm 0.49\%$, mean ± SD of five slices from five mice each, $P = 4.72 \times 10^{-2}$, $t_8 = 2.34$, Student's $t$-test). Taken together, we consider that our method is more suitable

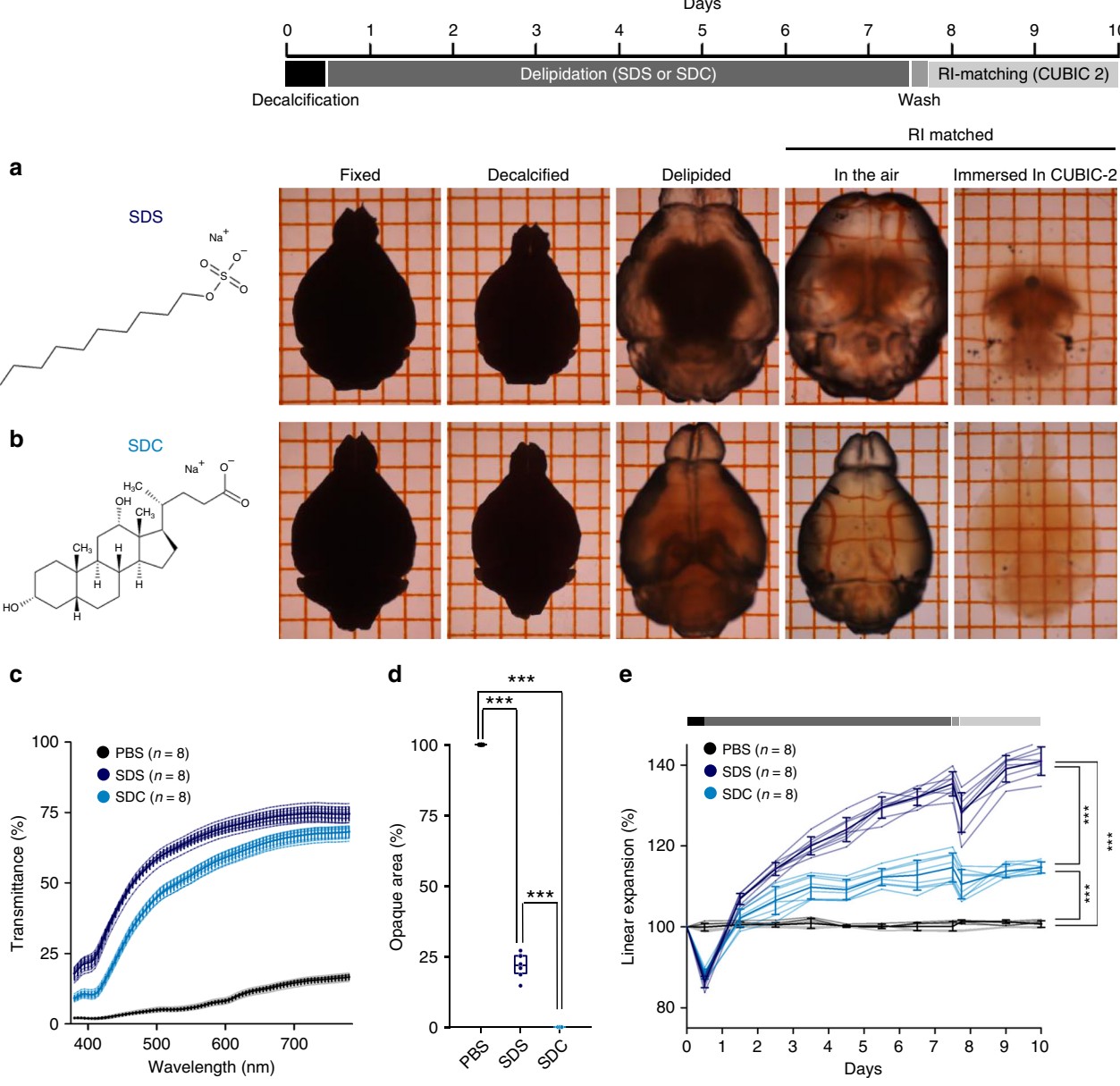

**Fig. 2 Rapid tissue clearing with sodium deoxycholate. a, b** Time-series of photographs of a brain taken at each step of tissue clearing. The fixed brain was treated with EDTA for 12 h followed by SDS **a** or SDC **b** for 7 d (daily refreshed), washed with PBS and refractive index (RI)-matched in Sca*le*CUBIC-2 for 2 d. In the SDS-treated group, the white matter remained opaque, and the sample swelled. In the SDC-treated group, the cleared brain was still pale yellow, but the white matter was transparent. The sample swelled to a lesser extent. The grid interval is 2 mm. **c** The light transmittance (380–780 nm) of fixed whole brains treated with SDS and Sca*le*CUBIC-2 (dark blue), with SDC and Sca*le*CUBIC-2 (light blue) or with PBS (black) ($n = 8$ brains each, $P = 2.22 \times 10^{-16}$, $F_{2,889} = 4.56 \times 10^{4}$, two-way ANOVA (between groups)). **d** Percentages of opaque area in the whole brains treated with SDS and Sca*le*CUBIC-2, with SDC and Sca*le*CUBIC-2, or with PBS ($n = 8$ brains each; box plots indicate median and 25−75% interquartile ranges, $P = 2.22 \times 10^{-16}$, $F_{2,21} = 4.08 \times 10^{3}$, one-way ANOVA; SDC versus SDS: ***$P = 7.47 \times 10^{-14}$, $Q_{2,21} = 26.73$; SDS versus PBS: ***$P = 4.36 \times 10^{-14}$, $Q_{2,21} = 94.78$; SDC versus PBS: ***$P = 4.36 \times 10^{-14}$, $Q_{2,21} = 121.5$, Tukey's test). **e** Time course of tissue expansion during clearing ($n = 8$ brains each, $P = 2.22 \times 10^{-16}$, $F_{2,21} = 6.66 \times 10^{2}$, one-way ANOVA; SDC versus SDS: ***$P = 4.41 \times 10^{-14}$, $Q_{2,21} = 33.15$; SDC versus PBS: ***$P = 9.92 \times 10^{-11}$, $Q_{2,21} = 17.69$; SDS versus PBS: ***$P = 4.36 \times 10^{-14}$, $Q_{2,21} = 50.84$, Tukey's test). Error bars represent standard deviations. Source data are provided as a Source Data file.

for analyzing the molecular and structural characteristics of cerebral vasculature.

**Vascular connectivity-retaining delipidation by a bile salt.** For the visualization of the 3D structure of vascular networks, conventional casting methods use corrosive chemicals and remove surrounding parenchymal tissues. We sought to replace this classical strategy with optical tissue clearing that retains

molecular information in the brain parenchyma. First, we conducted tissue clearing by delipidation with sodium dodecyl sulfate (SDS) for 1 week[5,13] followed by refractive index (RI) matching with Sca*le*CUBIC-2, an aqueous solution of sucrose, urea, and triethanolamine, for 2 d[24]. However, SDS-based delipidation insufficiently cleared the white matter of the brain and caused the specimens to become fragile and swell (Fig. 2a). We thus used sodium deoxycholate (SDC), a bile salt that has been used for clearing of plant tissues[25]. Although SDC has been considered

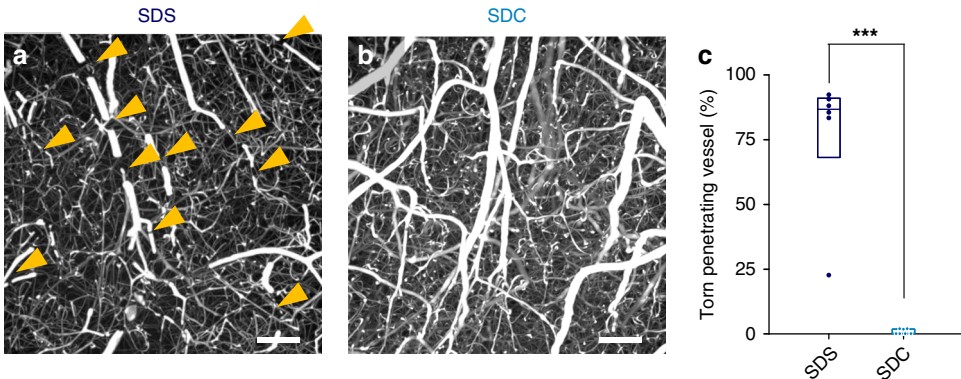

**Fig. 3 Breakage of the casted hydrogel is suppressed in SDC-based tissue clearing. a, b** Maximum projection of volumetric images of the neocortex. The vascular-casted samples were delipidated using SDS **a** or SDC **b** and were immersed in ScaleCUBIC-2. Some penetrating vessels were severed (arrowheads) in the SDS-treated group, whereas such breakages were rarely observed in the SDC-treated group. Scale bar = 100 μm. **c** The percentages of the broken penetrating vessels in the brain treated with SDS and SDC ($n = 6$ and 9 brains, respectively; box plots indicate median and 25−75% interquartile ranges, ***$P = 8.84 \times 10^{-7}$, $t_{13} = 8.70$, Student's $t$-test). Source data are provided as a Source Data file.

inapplicable for use in animal tissues[13,24], we found that it made the mouse brain transparent after 2-d RI matching with Scale-CUBIC-2, at least under our fixation condition (Fig. 2b). The average light-transmittance ratio of SDC-cleared tissues was slightly lower than that of SDS-cleared tissues (Fig. 2c, $n = 8$ brains each, $P = 2.22 \times 10^{-16}$, $F_{2,889} = 4.56 \times 10^4$, two-way ANOVA), but SDC-cleared tissues exhibited a spatially uniform RI across the specimens and had no opaque areas (Fig. 2d, $n = 8$ brains each, $P = 7.47 \times 10^{-14}$, $Q_{2,21} = 26.73$; SDC versus SDS, $P = 2.22 \times 10^{-16}$, $F_{2,21} = 4.08 \times 10^3$, Tukey's test after one-way ANOVA). Moreover, swelling was significantly reduced in SDC-cleared tissues (Fig. 2e, $n = 8$ brains each, $P = 4.41 \times 10^{-14}$, $Q_{2,21} = 33.15$ versus the SDS group, $P = 2.22 \times 10^{-16}$, $F_{2,21} = 666$, Tukey's test after one-way ANOVA). Thus, the connectivity of the vascular network was preserved in SDC-cleared tissues, whereas many neocortical penetrating vessels were fractured in SDS-cleared tissues due to overstretching of the fluorescent hydrogel (Fig. 3, $n = 6$ and 9 brains. $P = 8.84 \times 10^{-7}$, $t_{13} = 8.70$, Student's $t$-test). We refer herein to the combination of fluorescent Dex–GMA-based vascular casting and SDC-based tissue clearing as SeeNet (A method to See the cerebrovascular Network).

**Molecular phenotyping compatibility of SeeNet.** Next, we examined whether nucleic acids and proteins were maintained in SeeNet-treated brains. The vasculatures were cast using FITC–Dex–GMA, and the specimens were labeled using propidium iodide (PI), a DNA-staining reagent, or an antibody against Iba-1. Whole-mount imaging of the cerebral hemisphere revealed that PI and Iba-1 signals were detectable (Fig. 4a–f, $n = 7$ hemispheres each), suggesting that nucleic acids and proteins were retained in the tissues.

We also addressed the possible quenching of fluorescent proteins during the clearing procedure. We applied SeeNet to the brains of visually stimulated Arc-dVenus mice, in which active neurons express the modified yellow fluorescent protein dVe-nus[26]. Whole-mount imaging revealed that SeeNet retained the fluorescence of dVenus (Fig. 4g–i, $n = 7$ hemispheres).

We further examined the compatibility of SeeNet with other epitopes and fluorescent proteins using a light-sheet fluorescence microscope[7]. First, SeeNet-treated hemispheres were immunostained with fluorophore-conjugated antibodies against NeuN and GFAP (Supplementary Fig. 7). The optical sections of the stained samples were virtually indistinguishable from images of 2D-stained samples (Supplementary Figs. 5a, b and 7c, f), indicating that SeeNet-treated samples can be immunostained using these antibodies. Next, SeeNet protocol was applied to CX3CR1-GFP mice[27], which expresses GFP in microglia, and H-I7-iCre-imCherry mice[28], which expresses mCherry in a subpopulation of olfactory sensory neurons (Supplementary Fig. 8a–f). In CX3CR1-GFP mice, GFP-positive cells were observed throughout the brain. In H-I7-iCre-imCherry mice, the convergence pattern of mCherry-positive axons were clearly visualized with SeeNet protocol. These patterns were consistent with the expression patterns reported previously in these mouse lines, suggesting that the SeeNet protocol preserved the innate fluorescence of GFP and mCherry. In addition, SeeNet was able to visualize fluorescent protein expressed by intracranially injected adeno-associated virus (AAV), without apparent leakages of FITC–Dex–GMA around the injected sites (Supplementary Fig. 8g–i).

We next examined whether the fine subcellular structures of glial cells were preserved in the SeeNet protocol. GFAP-stained or CX3CR1-GFP-expressing SeeNet-treated brains were imaged by a confocal microscope. We successfully observed typical glial subcellular morphologies, i.e., astrocytic endfeet wrapping penetrating vessels[29] and microglial protrusions that likely touched capillaries[30] (Supplementary Fig. 9).

Then, we compared the SN ratio of 3D imaging data obtained from the SeeNet-treated samples to that obtained from samples prepared by a conventional protocol, intravenous injection of Texas Red-conjugated lectin and optical clearing with SDS/ScaleCUBIC-2 (Supplementary Fig. 10). Signals in SeeNet-treated brains were detectable at depths of at least 1000 μm, whereas signals in lectin/SDS/ScaleCUBIC-2-treated brains were almost indistinguishable from the background (Supplementary Fig. 10a, b, $n = 7$, 9 mice, respectively, $P = 2.22 \times 10^{-16}$, $F_{1,140} = 5.55 \times 10^2$; two-way ANOVA). The mean diameter of cerebral capillaries in SeeNet-treated samples was $6.58 \pm 1.21$ μm (Supplementary Fig. 10c, mean ± SD of 90 capillaries from 9 mice). This value was slightly but not significantly larger than the previously reported in vivo diameter ($6.3 \pm 1.1$ μm)[31], possibly due to changes in tissue volume caused by PFA fixation, delipidation, and RI-matching.

As the capillary diameters in SeeNet-treated samples were comparable to the pixel resolution of our macro-zoom light-sheet microscope, in which one pixel measures $6.45 \times 6.45$ μm², we attempted whole-brain imaging of cerebral vasculature at single-capillary resolution. Because of the high SN ratios of SeeNet-treated brain, we were able to visualize blood vessels in both gray

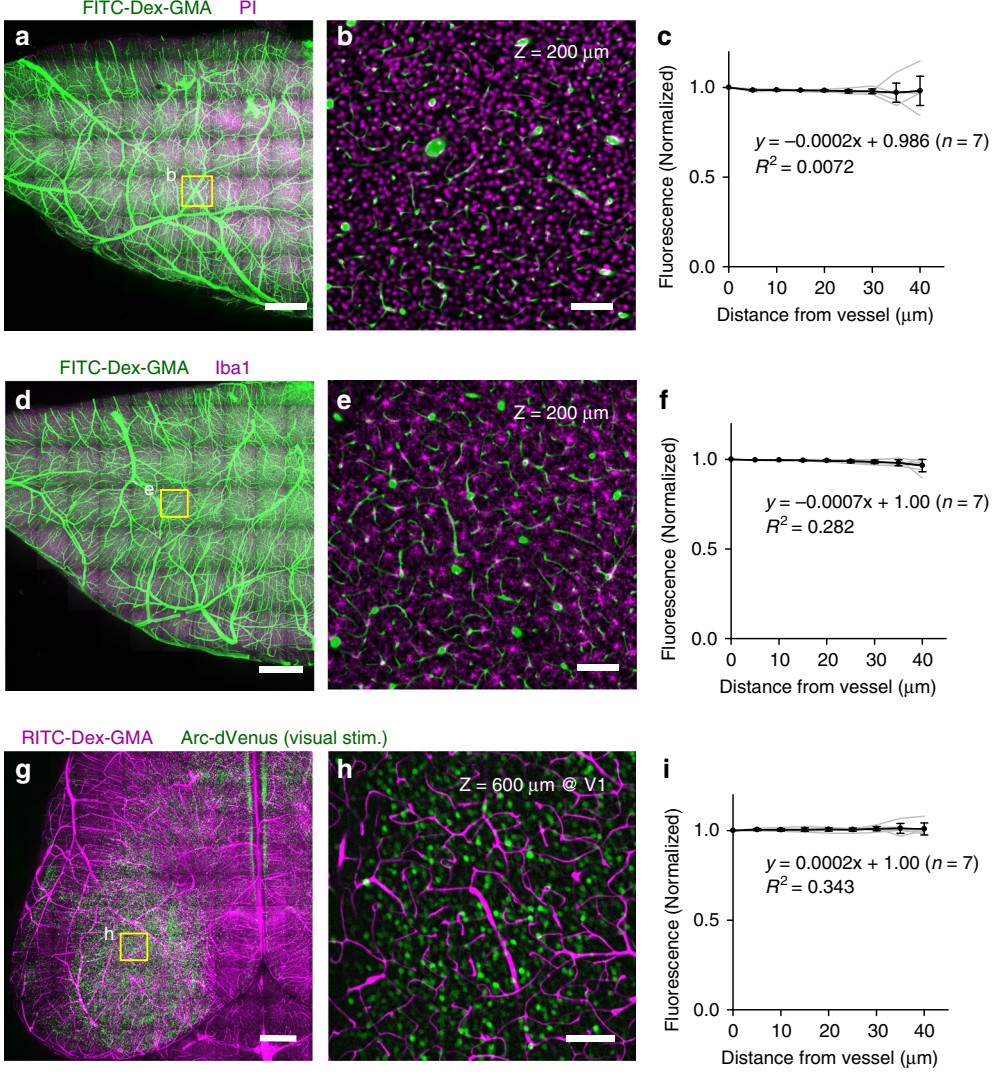

**Fig. 4 SeeNet retains nucleic acids, proteins, and fluorescence of endogenous probes. a** Maximum projection of volumetric images of the cerebral hemisphere. The vasculature was cast using FITC–Dex–GMA (green). After delipidation with SDC, nucleic acids were labeled using PI (magenta). Then, the sample was RI-matched using Sca*l*eCUBIC-2. Scale bar = 1 mm. **b** An optical section of a boxed region in panel **a** at $Z = 200$ μm. PI fluorescence was uniformly retained across the section. Scale bar = 100 μm. **c** The normalized median intensities of PI fluorescence were plotted as a function of the distance from the nearest vessel. The intensities were almost unchanged depending on the distance. $n = 7$ hemispheres from seven mice. **d–f** The same as panels **a–c** showing the immunohistochemical signal of anti-Iba-1 (magenta). **g–i** The same as panels **a–c** showing RITC–Dex–GMA (magenta) and Arc-dVenus (green). Error bars represent standard deviations. Source data are provided as a Source Data file.

and white matters of the brains with single-capillary resolution (Fig. 5a–d, Supplementary Movies 1 and 2). The brains were also immunostained using a FITC-conjugated antibody against α-smooth muscle actin (αSMA), a marker for arteries and arterioles. SeeNet could visualize the whole cerebral vasculature with molecularly identified arterioles and venules (Fig. 5a).

We also examined the SN ratio of 3D imaging data obtained from samples prepared with intravenous injection of an Alexa 647-conjugated anti-CD31 antibody and with optical clearing by ethanol and ethyl cinnamate (ECi)[32]. One sample was prepared from CX3CR1-GFP mice for the simultaneous examination of the fluorescence preservation of this method (Supplementary Fig. 11a–c). GFP fluorescence was barely visible in the gray matter of the brains and was almost invisible in the white matter (Supplementary Fig. 11d–f). The signal of endothelial cells was detectable in the gray matter but indistinguishable from the background noise in the white matter (Supplementary Fig. 11g–i). The SN ratio of the CD31 signal was significantly lower than that

of casted vessels in SeeNet (Supplementary Fig. 11j, k. $n = 4$ and 5 optical slices from 4 or 5 mice, respectively. Gray matter; ***$P = 1.32 \times 10^{-6}$, $t_7 = 15.1$, White matter; ***$P = 2.53 \times 10^{-5}$, $t_7 = 9.75$, Student's *t*-test).

**A microvascular path revealed by SeeNet**. Using SeeNet, we serendipitously discovered a previously unknown microvascular path. In rodents, the vascular paths of the hippocampus and the cerebral cortex were examined at the level of arterioles and venules. Once cerebral arteries are separated into the posterior cerebral artery and the longitudinal hippocampal artery, they are generally assumed to form independent networks until they converge in the internal jugular vein[1,33]. However, when we inspected SeeNet-treated samples at the single-capillary level, we found some microvessels that directly connected cortical and hippocampal vascular networks (Fig. 5e, Supplementary Movies 3, 4). We analyzed 54 arterioles and 57 venules of the

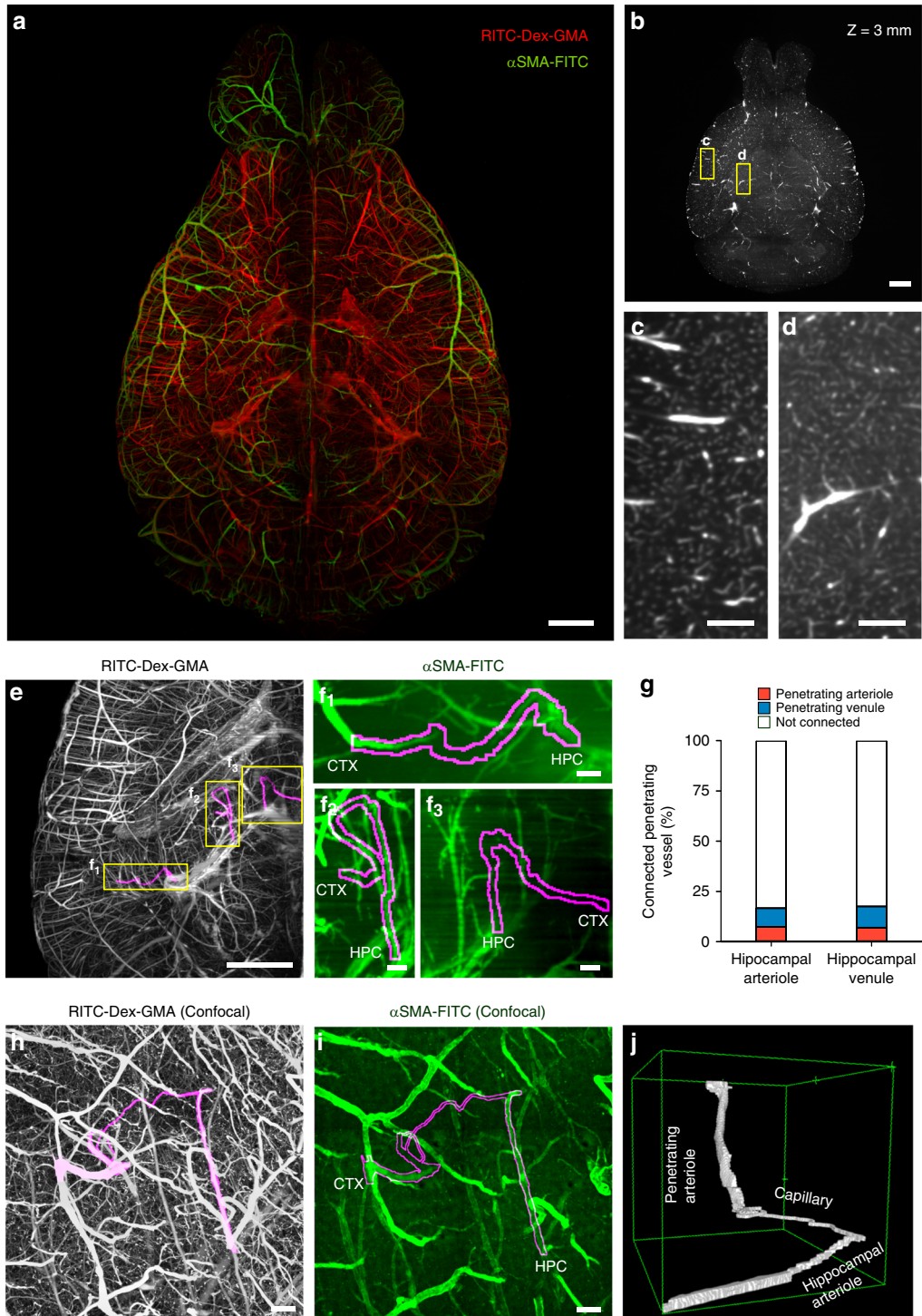

**Fig. 5 Large-scale molecular phenotyping-compatible 3D imaging of cerebral vasculature. a** Maximum projection of light-sheet fluorescence microscopic images of the entire vasculature in a SeeNet-treated brain (1.6× zoom, ×0.63 objective lens, numerical aperture: 0.15, working distance: 87 mm) into a single stacked photo. Casted vessels are shown in red. Arterioles immunolabeled with anti-αSMA are shown in green. Scale bar = 1 mm. **b** The RITC–Dex–GMA signal in **a**, shown on a grayscale, was optically sectioned at $Z = 3$ mm. Scale bar = 1 mm. **c, d** Magnified images of the boxed region of **b**. Vasculature in the gray matter **c** and white matter **d** was visualized with capillary resolution. Scale bar = 200 μm. **e** The RITC–Dex–GMA signal in a posterior part of the left hemisphere in **a** is shown on a grayscale. Magenta indicates cortico-hippocampal vascular paths observed in this area. Scale bar = 1 mm. See also Supplementary Movie 3. **f** The anti-αSMA immunosignal of the boxed regions in **e** is shown on a green scale. Magenta contours indicate cortico-hippocampal vascular paths. Each end of the vascular path is denoted by CTX (cortical side) or HPC (hippocampal side). Arteriole–arteriole ($f_1$), venule–arteriole ($f_2$), and venule–venule ($f_3$) connections were observed. Scale bar = 100 μm. See also Supplementary Movie 4. **g** The percentage of the connected pairs of dorsal hippocampal transverse arterioles (red) and venules (blue) to the total number of vessels observed. **h** Maximum projection of confocal images of casted vasculatures from the pia to the hippocampus. A cortico-hippocampal vascular path is shown in magenta. Scale bar = 100 μm. See also Supplementary Movie 5. **i** The anti-αSMA immunosignal in the same field is shown on a green scale. Scale bar = 100 μm. See also Supplementary Movie 6. **j** A 3D-rendered view of the cortico-hippocampal vascular path. Source data are provided as a Source Data file.

dorsal hippocampal transverse vessels from three mice (Fig. 5f, g). We found that 16.7% of the arterioles and 17.5% of the venules were connected to cortical penetrating vessels, suggesting that the flows of blood were not unidirectional. Interestingly, there were not only arteriole–venule connections (five hippocampal arteriole–cortical venule connections and four hippocampal venule–cortical arteriole connections) but also arteriole–arteriole and venule–venule connections (four and six connections, respectively). We confirmed, using higher resolution confocal imaging, that there is a cortico-hippocampal microvascular pathway connecting αSMA-immunopositive penetrating vessels and transverse vessels (Fig. 5h–j, Supplementary Movies 5, 6). These bypasses may actively exchange arterial blood and serve as a backup circulation pathway in the case of cerebral infarction[34].

## Discussion

In this study, we developed SeeNet, a method for the structural and molecular phenotyping of the 3D cerebral vasculature. The BBB-impermeable fluorescent crosslinker RITC–Dex–GMA enabled near-complete vascular casting with high SN ratios, and SDC-enabled relatively rapid (~10 d) tissue clearing without significantly affecting the synthesized fluorescent hydrogels or intrinsic nucleic acids or proteins.

Several pioneering tissue-clearing studies have attempted whole-brain visualization of cerebral vasculature using conventional vascular-labeling dyes[17,35,36]. However, due to the low SN ratios of the conventional dyes, these methods produce numerous false negatives. Although Di Giovanna et al. addressed this matter using temperature-dependent gelation of gelatin and FITC-conjugated albumin[10,14], the gelatin-based method has another source of false negatives because gelatin occasionally becomes clogged in microvessels. Moreover, when samples prepared by this protocol are molecularly characterized, further caution is necessary because antigenicity may be altered. In addition, the gelatin–FITC–albumin gel perfusion method costs > $200 per mouse and takes >6 weeks, making its use for routine vascular labeling difficult. To reduce false negatives, the use of transgenic mice is an alternative option. However, because the promoter for endothelial cells is heterogeneously active, particularly in adult mice[37], Cre-based double transgenic mice are required for complete labeling of cerebral vasculatures[18,20]. Therefore, the supply of double-transgenic mice is a bottleneck for research, and the use of double transgenic mice makes exploiting other transgenic mice difficult. In terms of these issues, SeeNet is quick, less costly and easily adoptable to wild-type and transgenic mice. SeeNet will enable the routinization of fine large-scale 3D tracing for molecularly characterizable cerebral vasculatures, thus advancing our understanding of the structural and functional organizations of cerebrovascular networks.

## Methods

**Animal ethics.** Two- to five-month-old C57BL/6J (Japan SLC, Inc., Japan), CX3CR1 -GFP (Stock No. 005582, The Jackson Laboratory)[27], H-I7-iCre-imCherry (Accession No. CDB0537T, RIKEN LARGE)[28], and Arc-dVenus transgenic mice[26] were used in this study. All animals were housed under a 12:12-h dark–light cycle (light from 07:00 to 19:00) at 22 ± 1 °C with ad libitum access to food and water. Animal experiments were performed with the approval of the Animal Experiment Ethics Committee at the University of Tokyo (approval number: 24-70) and according to the University of Tokyo guidelines for the Care and Use of Laboratory Animals. All experimental protocols were carried out in accordance with the Fundamental Guidelines for Proper Conduct of Animal Experiment and Related Activities in Academic Research Institutions (Ministry of Education, Culture, Sports, Science and Technology, Notice No. 71 of 2006), the Standards for Breeding and Housing of and Pain Alleviation for Experimental Animals (Ministry of the Environment, Notice No. 88 of 2006) and the Guidelines on the Method of Animal Disposal (Prime Minister's Office, Notice No. 40 of 1995).

**Synthesis of RITC-Dex-GMA or FITC-Dex-GMA.** The reactions are summarized in Supplementary Fig. 1. Step 1 (Supplementary Fig. 1a): A total of 4.0 g dextran 40 (049-22331, Wako Pure Chemical Industries Ltd., Japan) or dextran 200 (043-22611, Wako Pure Chemical Industries Ltd.) was dissolved in 36 ml DMSO (13406-55, Nacalai Tesque Inc., Japan) in a 50-ml conical tube under a nitrogen atmosphere. After dissolution of 0.8 g 4-dimethylaminopyridine (12922-02, Nacalai Tesque Inc.), 2.0 g GMA (17107-55, Nacalai Tesque Inc.) was added. The solution was gently shaken at 37 °C for 72 h. The reaction was stopped by adding an equimolar amount of concentrated HCl (18320-15, Nacalai Tesque Inc.) to neutralize 4-dimethylaminopyridine. The reaction mixture was precipitated by adding 4 ml of 3 M NaOAc (192-01075, Wako Pure Chemical Industries Ltd.) and 28 ml of isopropanol (29113-95, Nacalai Tesque Inc.). Then, the resulting fluffy product Dex–GMA was centrifuged at 6000 × g at 4 °C for 15 min. The precipitates were further washed in 5 ml of 70% EtOH (14713-95, Nacalai Tesque Inc.) and centrifuged again at 6000 × g at 4 °C for 15 min. Step 2 (Supplementary Fig. 1b): The resulting Dex–GMA was dissolved in 40 ml of DMSO containing a few drops of pyridine (162-05313, Wako Pure Chemical Industries Ltd.). Forty milligrams of RITC (R1755, Sigma-Aldrich, St. Louis, MO, USA) or FITC (7250, Sigma-Aldrich) was added, followed by the addition of 80 mg of dibutyltin dilaurate (D0303, Tokyo Chemical Industry Co., LTD., Japan). The mixture was heated at 98 °C for 3 h while it was stirred. The reaction mixture was cooled at room temperature and precipitated by adding 4 ml of 3 M NaOAc and 28 ml of isopropanol. Then, the resulting fluffy product RITC–Dex–GMA or FITC–Dex–GMA was centrifuged at 6000 × g at 4 °C for 15 min. The precipitate was further washed with 5 ml of 70% EtOH and centrifuged again at 6000 × g at 4 °C for 15 min. The product was stored at −20 °C. At the time of use, the product was dissolved in ~80 ml of PBS to form a 5% RITC/FITC-Dex–GMA solution.

**Vascular casting with RITC-Dex-GMA or FITC-Dex-GMA.** A mixture of 5% RITC–Dex–GMA, 5% acrylamide (011-08015, Wako Pure Chemical Industries Ltd.), and 0.25% VA-044 (017-19362, Wako Pure Chemical Industries Ltd.) was prepared in PBS on ice. We usually prepared 10 ml of the mixture containing dextran 40-based RITC–Dex–GMA and another 10 ml of the mixture containing dextran 200-based RITC–Dex–GMA. The mixtures were separately filtered through Whatman® qualitative filter papers (Grade 1, 1001 125, Sigma-Aldrich) that had been premoistened with PBS. Note that elevation in the temperature, degassing, and lack of filtration may lead to insufficient casting. Then, mice were anesthetized with 25 mg/kg pentobarbital (SOM04-YO1706, Kyoritsu Seiyaku Corporation, Japan) and 10 mg/kg xylazine hydrochloride (593-12891, Wako Pure Chemical Industries Ltd.). Using a peristaltic pump (Minipuls 3, Gilson, Middleton, WI), the mice were perfused with 20 ml of PBS containing heparin (10 U/ml, 17513-41, Nacalai Tesque Inc.) and verapamil (0.2 mg/ml, 222-00781, Wako Pure Chemical Industries Ltd.), 20 ml of 4% PFA, 10 ml of 5% RITC–Dex–GMA (dextran 40 based), 5% acrylamide, and 0.25% VA-044 and 10 ml of 5% RITC–Dex–GMA (dextran 200 based), 5% acrylamide and 0.25% VA-044 at 4 °C at 4 ml/min in this order. During perfusion, all perfusate-containing equipment was placed on ice. After the perfusion was complete, the mice were transferred into 50-ml conical tubes with their head down. Then, the tube was degassed to replace the air with nitrogen because oxygen in the air impedes hydrogel formation. The tube was submerged in a 37 °C water bath and incubated for exactly 3 h (longer incubation would make delipidation slower). Vascular casting by FITC–Dex–GMA was carried out in the same manner.

**Vascular labeling with fluorophore-conjugated lectin.** Mice were anesthetized with 25 mg/kg pentobarbital (SOM04-YO1706, Kyoritsu Seiyaku Corporation) and 10 mg/kg xylazine hydrochloride (593-12891, Wako Pure Chemical Industries Ltd.). For labeling by intravenous injection, the tail of the mouse was placed in a beaker filled with warm water at 40 °C for 30 s so that the tail veins became dilated. One hundred microliters of Texas Red®-conjugated lectin (Texas Red®-labeled Lycopersicon Esculentum lectin, TL-1176, Vector Laboratories, Burlingame, CA) or DyLight594-conjugated lectin (DyLight® 594 labeled Lycopersicon Esculentum Lectin, TL-1177, Vector Laboratories) was slowly injected via a lateral tail vein with a 27-G needle. More than 5 min after injection of lectin, animals were perfused with 20 ml of heparinized PBS, 20 ml of PFA, and 20 ml of acrylamide and VA044. For labeling via transcardial perfusion, anesthetized mice were perfused with 20 ml of heparinized PBS, 20 ml of PBS containing 100 μl of Texas Red-conjugated lectin, 20 ml of PFA, and 20 ml of acrylamide and VA-044.

**Vascular labeling with gelatin and FITC-albumin.** A gel solution containing 2% porcine skin gelatin type A (G1890, Sigma-Aldrich) was prepared in boiled PBS and cooled to <50 °C. The solution was then combined with 1% FITC-labeled albumin (A9771, Sigma-Aldrich), maintained with stirring at 40 °C, and filtered through Whatman® qualitative filter papers (Grade 1) that had been premoistened with PBS. The filtered solution was maintained at 40 °C before the perfusion. Then, mice were anesthetized with 25 mg/kg pentobarbital and 10 mg/kg xylazine hydrochloride. Using a peristaltic pump, the mice were perfused with 60 ml of PBS containing heparin (10 U/ml, 17513-41, Nacalai Tesque Inc.) and verapamil (0.2 mg/ml, 222-00781, Wako Pure Chemical Industries Ltd.), and 60 ml of 4% PFA. All solutions were maintained at 40 °C. After the perfusion of PFA, 20 ml of the

fluorescent gel perfusate was manually perfused at ~0.6 ml/s, with the body of the mouse tilted by 30° and the head down. The body was submerged in ice water to rapidly cool and solidify the gel as the final portion of the gel perfusate was pushed through. The brain was carefully extracted after ~15 min of cooling and submerged in 4% PFA at 4 °C for 12–16 h.

**Clearing protocol**. To minimize potential damage to the pial vessels, the brains of the fixed samples were removed together with their skulls. Then, the brains were transferred to 15-ml conical tubes containing 10 ml of 0.5 M EDTA (pH 8.0, 15130-95, Nacalai Tesque Inc.). The samples were decalcified overnight at 37 °C while shaking. Then, the softened skulls were gently removed from the shrunken brains. The brain-hydrogel matrices were transferred into 50-ml conical tubes containing 20 ml of 10% SDC (194-08311, Wako Pure Chemical Industries Ltd.) or 8% SDS (30400-85, Nacalai Tesque Inc.) in 200 mM boric acid (021-02195, Wako Pure Chemical Industries Ltd.)-based buffer at pH 8.5. Then, the samples were incubated for 7–10 d at 37 °C while shaking. The buffers were refreshed daily or every 2 d. The delipidated samples were washed three times in PBS at room temperature for 2 h each and were placed in 20 ml of Sca*le*CUBIC-2 at 37 °C for 2 d. Sca*le*CUBIC-2 was prepared as a mixture of 50 wt% sucrose (30403-55, Nacalai Tesque Inc.), 25 wt% urea (35907-15, Nacalai Tesque Inc.), and 10 wt% 2,20,20'-nitrilotriethanol (145-05605, Wako Pure Chemical Industries Ltd.).

**Vascular visualization with anti-CD31 and EtOH-ECi**. Two- to five-month-old CX3CR1-GFP or wild-type mice were anesthetized with 25 mg/kg pentobarbital and 10 mg/kg xylazine hydrochloride. Then, mice were treated with intravenous injection of 10 μg anti-CD31-Alexa 647 (102416; BioLegend, San Diego, CA, USA) in 100 μl saline solution. After 10 min, the mice were perfused using a peristaltic pump with 20 ml of PBS containing heparin (10 U/ml) and verapamil (0.2 mg/ml), and 20 ml of 4% PFA. After perfusion, the brain or the liver was removed and post-fixed with 4% PFA at 4 °C overnight. The brains were dehydrated in a graded ethanol (14712-05, Nacalai Tesque Inc.) series of 50% (24 h), 70% (24 h), and 100% (24 h, twice) at 4 °C with gentle shaking. The pH of EtOH was adjusted to 9.0 using trimethylamine (T0424, Tokyo Chemical Industry Co., LTD.). After dehydration, the samples were transferred to ECi (BD129295-500g; BLD Pharmatech Ltd., Shanghai, China) and incubated with gently shaking at room temperature until they became transparent (4–5 h).

**2D immunostaining**. The fixed samples were immersed in PBS containing 15% sucrose at 4 °C overnight and then in PBS containing 30% sucrose at 4 °C for 2 d. The samples were immersed in optimum cutting temperature (O.C.T.) compound and were frozen with dry ice. Sagittal sections at a 40-μm thickness were prepared at −20 °C using a cryostat (HM520, Thermo Fisher Scientific, Waltham, MA, USA). Immunohistochemistry was conducted on glass slides. Slices were rinsed in PBS at room temperature for 15 min and then permeabilized at room temperature for 30 min in 0.1 M phosphate buffer and 0.3% Triton X-100 (35501-15, Nacalai Tesque Inc.) with 10% goat serum (S-1000, Vector Laboratories). The samples were subsequently incubated with rat anti-CD31 (1:300; 550274, BD Biosciences, Franklin Lakes, NJ, USA), rabbit anti-NeuN (1:300; ABN-78, Sigma-Aldrich), rabbit anti-GFAP (1:300; Z0334, Agilent Technologies, Santa Clara, CA), or rabbit-anti-Iba1 (1:300; 019-19741, Wako Pure Chemical Industries Ltd) in 0.1 M phosphate buffer and 0.3% Triton X-100 with 10% goat serum at room temperature overnight. After the samples were rinsed in 0.1 M phosphate buffer for 3–5 min, they were incubated with the secondary antibody Alexa 488-labeled or Alexa 594-labeled goat anti-rat IgG (1:300; A-11006 or A-11007, Thermo Fisher Scientific) or Alexa 594-labeled donkey anti-rabbit IgG (1:300; R-37119, Thermo Fisher Scientific) in 0.1 M phosphate buffer with 10% goat serum at room temperature for 4 h. After the samples were rinsed with 0.1 M phosphate buffer for 3 × 10 min, they were mounted using VECTASHIELD Antifade Mounting Medium (H-1000, Vector Laboratories).

**3D immunostaining and nuclear staining**. Delipidated samples were immunostained using rabbit anti-Iba1 (1:200; 019-19741, Wako Pure Chemical Industries Ltd.), Alexa 488-conjugated mouse anti-NeuN (1:100; MAB377X, Merck-Millipore, Burlington, MA, USA), Cy3-conjugated mouse anti-GFAP (1:100; C9205, Sigma-Aldrich), or FITC-conjugated mouse anti-αSMA (1:100; F3777, Sigma-Aldrich) in 0.5% (v/v) Triton X-100, 0.25% casein (12922-02, Nacalai Tesque Inc.), and 0.05% sodium azide (31208-82, Nacalai Tesque Inc.) at 37 °C for 6 d (anti-Iba1) or 15 d (anti-NeuN, anti-GFAP, anti-αSMA) while shaking. The total volume of the staining solution was 800 μl for the whole brain and 400 μl for the hemisphere. Anti-Iba1-treated samples were then washed three times in PBS for 2 h each and treated with Alexa 594-labeled goat anti-rabbit IgG (1:300; A-11012, Thermo Fisher Scientific) at 37 °C for 6 d with shaking. The stained samples were washed three times in PBS at room temperature for 2 h each and immersed in Sca*le*CUBIC-2. Except for anti-αSMA-stained samples, immunostained samples were refixed with 1% formaldehyde (16223-55, Nacalai Tesque Inc.) at 25 °C for 24 h before the RI matching with Sca*le*CUBIC-2. For nuclear staining, the delipidated samples were immersed in PBS containing 5 μg/ml PI (P3566, Thermo Fisher Scientific), 0.5% (v/v) Triton X-100, and 0.05% sodium azide at 37 °C for 12 d with shaking. The solution was refreshed every 3 d.

**AAV production**. pAAV-CaMKIIa-mCherry-WPRE-hGH-polyA was constructed from pAAV-CaMKIIa-hM3D(Gq)-mCherry-WPRE-hGH-polyA (Addgene 50476) using *Sal*I (1080 A; TaKaRa Bio Inc., Japan), *Eco*RV (1042A; TaKaRa Bio Inc.), and the In-Fusion HD Cloning Kit (Z9648N; TaKaRa Bio Inc.). Recombinant AAV was generated by triple transfection of the 293 AAV cell line (AAV-100; Cell Biolabs, Inc.) with AAVdj rep-cap and pHelper from the AAV-DJ Helper Free Packaging System (VPK-400-DJ; Cell Biolabs, Inc., San Diego, CA) and pAAV-CaMKIIa-mCherry-WPRE-hGH-polyA, using PEI-max (24765; Polysciences, Inc., Warrington, PA). AAV vectors were purified using the AAVpro Purification Kit All Serotypes (6666; Takara Bio Inc.). Virus titers were then determined by qPCR using the primer pair AAV2 ITR[38], Luna Universal® qPCR Master Mix (M3003S; New England Biolabs, Ipswitch, MA), and the LightCycler® qPCR 2.0 system (DX400; Roche, Basel, Switzerland).

**Virus injection**. Mice anesthetized with isoflurane (Pfizer Inc., New York, NY) were placed in a stereotaxic frame following subcutaneous administration of ropivacaine (100 μl; Aspen Japan, Japan). rAAVdj-CaMKIIa-mCherry (1.5 × 10^13 GC/ml, 250 nl) was pressure-injected into the primary somatosensory area (S1; caudal −1 mm, lateral ± 2.25 mm from the bregma, and ventral 0.5 mm from the surface of the brain) and the primary visual cortex (V1; caudal −3.5 mm, lateral ± 2.5 mm from the bregma, and ventral 0.5 mm from the surface of the brain) at a rate of 50 nl/min using a syringe pump (KD Scientific Inc., Holliston, MA, USA) connected to glass pipettes (30-0034; Harvard Apparatus, Holliston, MA). After intraperitoneal administration of meloxicam (100 μl of 0.5 mg/ml; Boehringer Ingelheim Vetmedica Inc., Duluth, GA) at the end of surgery, the mice were returned to their home cages. SeeNet was conducted 2 weeks after the virus injection.

**Light transmittance measurements**. Vascular casting of wild-type mice was performed with Dex–GMA (nonfluorescent), acrylamide, and VA-044. Then, the samples were decalcified, delipidated, and RI adjusted, as described above. Light transmittance was measured from 380 to 780 nm at 5-nm intervals with an integrating sphere (Spectral Haze Meter SH 7000, Nippon Denshoku Industries, Japan). The samples were placed at the center of the optical cell (2277, Nippon Denshoku Industries). Two values were measured by the instrument: diffused (scattered) light transmittance (%), which is the value of the light transmitted through the tissue, and parallel (nondiffused) light transmittance (%), which is the value of light detected without any prevention. The transmittance was defined as the diffused light transmittance divided by (100–parallel light transmittance)[7].

**Microscopy**. Images were acquired at a Z-step of 10 μm and *xy* resolution of 0.794 μm using an inverted confocal microscope (×10, NA: 0.4, WD: 2.17 mm; Cell-Voyager™ CV1000, Yokogawa, Japan) and 488 and 561-nm lasers. Whole-brain images were acquired using a custom-made light-sheet fluorescence microscope (developed by Olympus Corporation, Japan) using a ×0.63 objective lens (NA: 0.15, WD: 87 mm) and an sCMOS Camera (Neo 5.5, Andor Technology, Belfast, United Kingdom) with a 1.6× digital zoom (MVX10, Olympus Corporation). Lasers with wavelengths of 488, 560, and 639 nm were used for image acquisition. Images were captured at a Z-step of 7 μm with a pixel resolution of 6.45 μm.

**Quantification of the SN ratio and overlap ratio**. Each image was semi-automatically binarized. RITC–Dex–GMA images were preprocessed using the background subtraction algorithm and then binarized using the triangle method. Images of CD31 or Texas Red-conjugated lectin were preprocessed using the background subtraction algorithm, bandpass filtered, and binarized using Yen's algorithm because they could not be adequately binarized with the triangle method due to their weaker signals. Then, the binarized images were scrutinized by eye, and apparent false-positive signals were manually corrected. The overlap ratio was calculated as the quotient of the area where the CD31-positive pixels and the RITC–Dex–GMA or Texas Red-conjugate lectin-positive pixels were colocalized divided by the area of CD31-positive pixels. The SN ratio of the images was defined as the Z-score of the mean intensities of the signals in the RITC–Dex–GMA or Texas Red-conjugated lectin-positive areas calculated using the mean and the standard deviation of the signals in the RITC–Dex–GMA or Texas Red-conjugated lectin-negative areas in the tissue. The SN ratio of the optical sections of CD31-EtOH–ECi-treated or SeeNet (RITC–Dex–GMA)-treated samples was calculated in the same manner. Images were preprocessed using the background subtraction algorithm and then binarized using the triangle method. Region of interest (150 × 150 pixels) was defined in the gray matter or the white matter of the optical slices at Z = 3150 μm.

**Quantification of PI, Iba-1, and dVenus signals**. For the measurement of PI and Iba-1 signal intensities, vascular casting of the samples was performed with FITC–Dex–GMA, acrylamide, and VA-044. The hemispheres of the samples were delipidated, stained, and RI adjusted. Images were taken at 200 μm deep from the pia. The signals of the casted vessels were automatically binarized using the triangle algorithm, and the 2D Euclidean distances from the nearest vessels were calculated for every pixel using the Fiji plugin. Only a few pixels were more than 40 μm away from the casted vessels, and therefore, they were

excluded from the quantification. The signals of PI or Iba-1 were automatically binarized with the triangle algorithm, and the median intensities of the signals in PI-positive or Iba-1-positive pixels within a defined range of distances from the vessels were calculated. To measure dVenus signal intensities, Arc-dVenus transgenic mice were housed under dark conditions for at least 60 h and then exposed to an environment lined with alternating black and white stripes (2 cm in width) on the walls at a light level of ~1500 lx. After 5 h of exposure to the environment, vascular casting was performed using RITC–Dex–GMA, acrylamide, and VA-044. The whole brains were delipidated, stained, and RI adjusted. Images were taken from the deep layers of the primary visual cortex (~600 μm deep from the pia) for quantification.

**Quantification of tissue swelling and opaque areas**. Vascular casting of wild-type mice was performed using nonfluorescent Dex–GMA, acrylamide, and VA-044. To measure the initial sizes of the brains, they were removed from the skull before decalcification. Then, the brains were treated with EDTA, delipidated, and RI adjusted as described above. At each time point of quantification, the samples were placed on a glass dish with the grid on the bottom and then imaged using a microscope. The area of the brain was calculated in manually segmented images, and the linear expansion value was determined by the square root of the area size changes. Opaque areas were calculated in light-transmitted images of the optically cleared samples. Specifically, the cleared samples immersed in RI-matching solution were imaged, bandpass filtered, and automatically binarized using the triangle algorithm. The original values of the pixels that made the grids on the glass dish invisible in the binarized images were calculated. The areas composed of pixels with values lower than the minimum values calculated above were defined as opaque areas (the pixels on the grid were not included). Then, the opaque areas were divided by the areas of the cleared samples that were not immersed in the RI-matching solution. The areas of the olfactory bulbs were excluded from the calculation because the olfactory bulbs were usually lost during SDS-based clearing.

**Quantification of the breakage of blood vessels**. Wild-type mice underwent vascular casting. The hemispheres of the samples were delipidated using SDS or SDC and were immersed in Sca*le*CUBIC-2. Images of the neocortex were captured using a confocal microscope. All identifiable penetrating vessels in a defined area (~50–60 vessels per sample) were inspected by eye, and breaks in the vessels were manually counted. Penetrating vessels were chosen for quantification because their large size and straight morphology allowed us to estimate originally connected vessels with high confidence. Pial vessels were not analyzed because they are more vulnerable during sample handling, and a portion of pial venules could be peeled off along with the dural venus sinuses. Furthermore, quantitative comparisons of the breakage of pial vessels were impractical because the majority of pial vessels were lost in SDS-treated samples.

**Statistical analysis**. Student's $t$ test was used for comparisons between two groups. One-way or two-way analysis of variance, Kruskal–Wallis test, Tukey's test, and Steel–Dwass test were used for comparisons among more than two independent groups. All tests were two-sided. Statistical significance was set at $P < 0.05$.

**Reporting summary**. Further information on research design is available in the Nature Research Reporting Summary linked to this article.

## Data availability
Source data underlying Figs. 1–5 is available as a Source Data file. Any additional data are available from the corresponding author upon reasonable request.

## Code availability
Any codes used in this study are available from the corresponding author upon reasonable request.

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

## Acknowledgements

We thank Olympus Engineering for helping with the microscope design. We thank Ryuta Koyama and Megumi Andoh for housing CX3CR1-GFP mice and Yu Sato, Kenichi Makino and Sakura Okada for preparing materials. This work was supported by JST ERATO (JPMJER1801 to Y.I.), JSPS Grants-in-Aid for Scientific Research (18H05525 to Y.I.), the Human Frontier Science Program (RGP0019/2016 to Y.I.), AMED-CREST (AMED/MEXT to H.R.U.) and CREST (JST/MEXT to H.R.U.); Brain/MINDS (AMED/ MEXT to H.R.U.); the Basic Science and Platform Technology Program for Innovative Biological Medicine (AMED/MEXT to H.R.U.); a Grant-in-Aid for Scientific Research (S) (JSPS KAKENHI grant 18H05270 to H.R.U.); PRESTO from JST (JPMJPR15F4 to E. A.S.); a Grant-in-Aid for Young Scientists (A) (JSPS KAKENHI grant no. 15H05650 to E. A.S.); Grants-in-Aid for Scientific Research on Innovative Areas (JSPS KAKENHI grant 17H06328 to E.A.S.); Grants-in-Aid for Challenging Research (Exploratory) (JSPS KAKENHI grant 18K19419 to E.A.S.); Grants-in-Aid from the Takeda Science Foundation (H.R.U. and E.A.S.); and a Grant-in-Aid from the Japan Foundation for Applied Enzymology (E.A.S.). This work was conducted partly as a program at the International Research Center for Neurointelligence (WPI-IRCN) of The University of Tokyo Institutes for Advanced Study at The University of Tokyo.

## Author contributions

T.M. conceived and designed the study. T.M. performed all the experiments and the analysis, except the intracranial stereotaxic injection of AAV, which was performed by S.M. E.A.S., and H.R.U. assisted imaging by light-sheet fluorescence microscope. S.Y. provided Arc-dVenus Tg mice. A.N. and H.T. provided H-I7-iCre-imCherry mice. T.M. and Y.I. wrote the manuscript. All authors discussed the results and commented on the manuscript text.

## Competing interests

Part of this study was performed in collaboration with the Olympus Corporation.
