## [Peer Review File · Nature Communications]

The paper by Miyawaki and coworkers presents a vascular casting method (SeeNet) suitable for optical clearing and subsequent fluorescent imaging of entire vascular networks. The authors claim, in some cases without pertinent support, that their method is superior to well established predecessors.

Comments bearing conceptual relevance.

In particular, the author claim to be superior in reproducibility compared to a FITC-based method that was used in tens of papers. It is not clear by which method the authors reached what seems to be a strongly biased conclusion (Supp Table 1).

There are contradictory results presented in the paper. In figure 3, the authors selected images that show that the hydrogel combined with SDC clearing results in no broken vessels (with a percent close to 0). Nevertheless, zooming into Figure 5a shows broken vessels:

This reviewer is concerned that this might indicate again a bias in the selection of region for the analysis presented in figure 3.

A critical point only partially covered in the paper is the preservation of endogenously expressed fluorescent proteins and endogenous epitopes. The author should have demonstrated that their method is compatible a wide range of FP, particularly GFP and its variants in neurons, showing that fine structures are not disturbed by their method. Not being able to preserve FP renders any clearing method almost irrelevant today. Moreover, claiming that epitopes are preserved cannot be based on a single case (IBA1, fig 4). The authors should have presented an extensive antibody survey covering different cell types and sub-cellular structures.

The intriguing observation regarding the vascular connection between cortical and hippocampal vascular network has to be largely quantified, this can be a good demonstration of the tool presented here.

Another concerning issue pointing to some biased selection of analysis regions is exemplified in Supp Fig 3. Notably, the staining of CD31 is not even as shown in the inset (d) of the said figure. Yet the FITC-albumin on the same region (as seen in panes a and c) appears complete which would have changed the statistic shown (3). To overcome this, we would recommend to inject IV Dylight488 in vivo and use it as a comparison with the vascular casting methods.

Page 4 - line 6-11, fig. 1b:

PFA disrupts physiological conditions and therefore causes micro changes in vascular outer and inner structures. Moreover perfusion in 4C (using polymer) may vasodilate vasculature (disruption in vascular morphology). The authors should have presented a comparison with vessel diameters measured in vivo.

Reviewers' comments:

Reviewer #2 (Remarks to the Author):

This novel method promises to be a very useful tool in the study of pathomechanisms of vascular dementia and small vessel disease. My only comment refers to whether this method would be amenable to be used for experimental studies using tracers injected into the different compartments of the brain. If so, perhaps the authors would consider adding a statement about the applications of this method- for example in the experimental studies of drainage of fluids from the brain.

Reviewer #3 (Remarks to the Author):

The manuscript “Full visualization and molecular characterization of whole-brain vascular networks with capillary resolution” by Miyawaki et al. describes a modified procedure using RITC-Dex-GMA for fluorescent visualization of blood vessels within the brain in whole-mount specimens. The authors describe a clearing technique using SDC previously used in plants. They compare their modified method with other established labeling techniques for brain vessels like injection with fluorescently labeled lectin or FITC-albumin-hydrogel perfusion. Their analyses include the quantification of signal-to-noise-ratios and co-localization with the endothelial marker CD31. In addition, they suggest a hitherto unknown microvascular path between the hippocampus and the cerebral cortex.

The data provided seems to be of good quality, the data and statistics were clearly explained and the supplementary material complements the main data in a reasonable way.

Regarding the information provided in the manuscript, the reviewer clearly questions the originality and significance in the field of vessel visualization and clearing techniques for light sheet microscopy:

1. The authors do not cite recent literature in which for example the 3DISCO clearing method in combination with gelatin-FITC-albumin hydrogel was successfully used in mouse brains (Lugo-Hernandez et al, 2017).
2. Furthermore, there are already other than the mentioned clearing procedures available which do not influence fluorescence signals as stated by the authors (Klingberg et al, 2017). In addition, the authors mention two references which should show that SDC was supposed not to be suitable for clearing in animal tissue (page 6, row 5-6), but these manuscripts do not mention SDC at all.
3. The authors statement that perfusing brains with gelatin and FITC-albumin would produce costs of 200 US\$/sample cannot be verified. The cost of all materials needed for this method averages out at around 30 US\$.
4. The manuscript mainly provides data obtained from projections of confocal microscope pictures in 2D immunostainings of cutted sections. This clearly counteracts the purpose of whole-organ preparation for 3D analysis. The data provided from light sheet microscopy is recorded in low magnification and therefore the hypothesis of hippocampal and cortex connecting vessels is demonstrated in poor resolution pictures as well as videos.

Taking into account the requirements of manuscripts to be published in Nature Communications the reviewer clearly recommends a rejection of this manuscript.

We are grateful for the very helpful and constructive comments from all the reviewers and have taken the advice to conduct additional experiments and revise our manuscript. We hope that these corrections now meet the reviewers' expectations. We thank all of you for the suggestions, which have substantially improved this paper. The revised parts are highlighted in blue in the manuscript.

Each specific comment is labeled with the heading of "Comment #...", and our response to the comment is shown in blue with the heading "Answer #..".

Reviewer #1 (Remarks to the Author):

We appreciate that this reviewer found scientific value in our manuscript. Because of these comments, we were pleased to be able to revise and improve the manuscript. Individual responses are listed below:

Comment #1.

In particular, the author claim to be superior in reproducibility compared to a FITC based method that was used in tens of papers. It is not clear by which method the authors reached what seems to be a strongly biased conclusion (Supp Table 1).

Answer #1

Thank you for raising this point. Tsai's paper stated "all vessels are filled in frontal cortex, parietal cortex (Figure 1, compare c, e, g, d, f, h), thalamus, striatum, and cerebellum". As shown in Figure S3f, we failed to reproduce the results. We admit that "bad reproducibility" was too strong of an expression and may potentially give readers the wrong impression. The "false negative" column in Supplementary Table 1 is enough to express our point, and thus, we have decided to remove the "reproducibility" column in Supplementary Table 1.

Comment #2.

There are contradictory results presented in the paper. In figure 3, the authors selected images that show that the hydrogel combined with SDC clearing results in no broken vessels (with a percent close to 0). Nevertheless, zooming into Figure 5a shows broken vessels: This reviewer is concerned that this might indicate again a bias in the selection of region for the analysis presented in figure 3.

Answer #2

We apologize for a lack of detailed explanation of these data. In Figure 3, we show that the percentage of broken penetrating vessels is “close to 0”. Indeed, we did not state that there were “no broken vessels”. The claim of this figure is that the breakage is suppressed in our SDC-based protocol compared to that in the SDS-based protocol. This fact does not contradict the reviewer’s observation that there are a few broken pial vessels. In addition, the pial vessels are located on the surface of samples and are more prone to breakage during sample handling (not due to swelling) than penetrating vessels. However, we think that quantitative comparison of the breakage of pial vessels is not very practical because pial vessels are largely lost in SDS-treated samples. To avoid confusion, we have detailed the rationale for quantification of penetrating vessels rather than pial vessels (in the Methods section, P24, L9-15).

Comment #3.

A critical point only partially covered in the paper is the preservation of endogenously expressed fluorescent proteins and endogenous epitopes. The author should have demonstrated that their method is compatible a wide range of FP, particularly GFP and its variants in neurons, showing that fine structures are not disturbed by their method.

Not being able to preserve FP renders any clearing method almost irrelevant today.

Answer #3

We agree. Demonstration of the preservation of florescent proteins will place our clearing method superior to several clearing methods that cannot preserve florescent proteins. First, we would like to emphasize that, in Figure 4g-i, we already presented the preservation of dVenus fluorescence in neurons using our method. dVenus is a derivative of YFP, which in turn is a derivative of GFP. According to the reviewer’s advice, we have applied our SeeNet protocol to GFP-expressing transgenic mice (CX3CR1-GFP mice), mCherry-expressing mice (H-I7-iCre-imCherry mice) and WT mice expressing mCherry through intracranial injection of AAV. As shown in the new Supplementary Figure 7, we found that the fluorescence of GFP and mCherry was preserved. The images have also shown that even axons of neurons could be visualized using our clearing protocol. Moreover, we have shown in Supplementary Figure 8d-f that microglial protrusions touched capillaries, indicating preservation of GFP signal in subcellular glial structures. Thanks to the comment of this reviewer, we are pleased to further demonstrate the potential usability of our method.

Comment #4-1

Moreover, claiming that epitopes are preserved cannot be based on a single case (IBA1, fig 4). The authors should have presented an extensive antibody survey covering different cell types and sub-cellular structures.

Answer #4-1

We agree. We should have presented the compatibility with antibodies that detect different markers of cells and subcellular structures. In Figure 5a, we already presented that another epitope, α SMA, as well as Iba1, was also stained in SeeNet-treated samples. According to the reviewer's suggestion, we stained for NeuN and GFAP, markers for neurons and astrocytes, respectively (Supplementary Figure 6). The results showed that immunohistochemical detection of these epitopes was feasible. We also found that astrocytic subcellular structures could be observed using SeeNet-treated samples (Supplementary Figure 8a-c).

Comment #4-2

The intriguing observation regarding the vascular connection between cortical and hippocampal vascular network has to be largely quantified, this can be a good demonstration of the tool presented here.

Answer #4-2

Thank you for your positive comment on our finding. We have now quantified the cortico-hippocampal microvascular pathways and their putative direction of blood flow in three mice (Figure 5e-g). Indeed, the bypasses were numerically few, and we would not be able to discover the rare cortico-hippocampal network without our clear method, which again indicates the potential usability of our method.

Comment #5

Another concerning issue pointing to some biased selection of analysis regions is exemplified in Supp Fig 3. Notably, the staining of CD31 is not even as shown in the inset (d) of the said figure. Yet the FITC-albumin on the same region (as seen in panes a and c) appears complete which would have changed the statistic shown (3). To overcome this, we would recommend to inject IV Dylight488 in vivo and use it as a comparison with the vascular casting methods.

Answer #5

Thank you for suggesting this line of experiments. Unfortunately, intravenous injection of

Dylight488, a dye that has no endothelial cell-binding motif and has the same emission spectrum as FITC, cannot be used as the positive control for the vasculature staining. In the first place, the statement in the legend of Supplementary Figure 4e (“(3)” should be written as “(e)”, i.e., previous Supplementary Figure 3e) was not about the overlap ratio between signals of the perfused dyes and anti-CD31 but rather the “percentages of CD31-immunopositive areas in the slices” prepared from gelatin-FITC-albumin-treated or FITC-Dex-GMA-treated samples. Moreover, this value was not derived from an arbitrarily selected region but from the whole slices. Our description was likely misleading in Supplementary Figure 4e, which seemed to be the same as Figure 1e, in which the value represents the signal overlap ratio of the perfused dyes and anti-CD31. Our main claim of Supplementary Figure 4 is that the staining of anti-CD31 became uneven in the FITC-albumin-treated sample, implying that the antigenicity of this epitope was affected in this protocol. To avoid confusion and further examine this possible caveat that should be taken into consideration when the FITC-albumin-based method is used, we have added a new Supplementary Figure 5, in which we have investigated the staining patterns of three other epitopes, NeuN, GFAP, and Iba1. All these epitopes exhibited deteriorated signal intensities and uneven staining patterns in the FITC-albumin-based method, whereas those in PFA-treated and Dex-GMA-treated samples were similar.

Comment #6

Page 4 - line 6-11, fig. 1b: PFA disrupts physiological conditions and therefore causes micro changes in vascular outer and inner structures. Moreover perfusion in 4C (using polymer) may vasodilate vasculature (disruption in vascular morphology). The authors should have presented a comparison with vessel diameters measured in vivo.

Answer #6

Histological studies usually need perfusion with PFA, but we agree that PFA and other manipulations may change the microscopic structures of brain tissues. Thus, this undesired effect should be examined in more detail. In the new Supplementary Figure 9c, we measured the diameters of capillaries in our brain samples. The mean diameter was $6.58 \pm 1.21 \mu\text{m}$, which seems to be slightly larger than the values previously reported in vivo ($6.3 \pm 1.1 \mu\text{m}$, according to Cruz Hernandez et al., 2019). This difference might be due to moderate swelling caused during tissue clearing. We have added the statement about this in P9 L20-25.

Reviewer #2

Comment #1

This novel method promises to be a very useful tool in the study of pathomechanisms of vascular dementia and small vessel disease. My only comment refers to whether this method would be amenable to be used for experimental studies using tracers injected into the different compartments of the brain. If so, perhaps the authors would consider adding a statement about the applications of this method- for example in the experimental studies of drainage of fluids from the brain.

Answer #1

Thank you for the positive evaluations, which have encouraged us to resubmit this manuscript. According to this comment, we have carefully discussed the potential usability of this protocol in conjunction with small molecule-based tracers (for example, smaller Dex-GMA with different colors), although we were not confident that it is indeed possible. Because perfusion with PBS, PFA, and monomers per se may affect the flux inside the brain parenchyma, the experimental results are difficult to interpret. However, inspired by this comment, we further examined the compatibility of our clearing method with viral tracers injected into the neocortex (Supplementary Figure 7g-i). The results showed no apparent leakage of the dyes, suggesting that SeeNet is compatible with injection experiments when mice are allowed to recover for at least 2 weeks. We would like to thank this reviewer for providing an opportunity to demonstrate another useful application of our protocol.

Reviewer #3:

We thank the reviewer for the constructive comments, which have greatly improved our work. Individual responses are provided below:

Comment #1.

The authors do not cite recent literature in which for example the 3DISCO clearing method in combination with gelatin-FITC-albumin hydrogel was successfully used in mouse brains (Lugo-Hernandez et al, 2017).

Answer #1

Thank you for your suggestion. We have included the citation in the revised manuscript. In the cited paper, the authors tested only the combination of gelatin FITC-albumin-based method and 3DISCO. These methods were cited in the previous manuscript, in which we noted that both the gelatin FITC-albumin-based method and 3DISCO had some drawbacks. In fact, the authors of

this paper did not quantify the staining efficiency and admitted that 3DISCO quenches fluorescent proteins by stating “Although 3 h incubation was suggested in the original 3DISCO protocols, in particular when working with fluorescent proteins, we found this insufficient for imaging of brain capillaries”. Moreover, the authors did not examine possible changes in antigenicity. Therefore, to emphasize the advancement of our method, we investigated the antigenicity of other epitopes (new Supplementary Figure 5) and compared the GFP-quenching effects of our clearing method with those of the EtOH-ECi-based clearing method, which was shown to preserve the fluorescence of GFP more than 3DISCO or BABB (Supplementary Figure 7a-c and 10d-f). The results showed that our method was superior in preserving the antigenicity as well as the fluorescence of fluorescent proteins. We believe that the completeness of vascular staining and preservation of antigenicity and fluorescent proteins are important factors for molecular and structural investigations of cerebral vasculature. Indeed, our method achieves both, and we would like these benefits to be re-evaluated.

Comment #2-1.

Furthermore, there are already other than the mentioned clearing procedures available which do not influence fluorescence signals as stated by the authors (Klingberg et al, 2017).

Answer #2

Thank you for this comment. The method of Klingberg et al. (2017) has not been tested in the brain. These authors were not able to make the kidney, which is smaller than the brain and does not contain lipid-rich white matter, transparent at the center of this organ (Klingberg et al., Figure 1). Blood vessels located near the tissue surface of the kidney were clearly visualized, but deep vessels were only obscurely visualized due to the low SN ratio of the dye and the efficiency of clearing (Klingberg et al., Figure 2). Furthermore, contrary to this reviewer’s comment, the authors noted that their protocol influences the fluorescence by stating “In general, we found a slow decay of the signal-to-background ratio of EYFP over time (Figure 1, A and B).” What these authors claimed is that their protocol preserves fluorescence of fluorescent proteins better than other organic solvent-based protocols, which are known to quench the fluorescence more than hydrophilic solvent-based protocols (Hama et al., 2015., Chen et al., 2017); please note that only one hydrophilic solvent protocol, SeeDB, was tested in their paper.

To examine the compatibility of Klingberg’s method in the brain and to compare its quenching effect with our clearing method, we applied Klingberg’s and our protocol to CX3CR1-GFP mice, which express GFP in microglia (new Supplementary Figure 7a-c and 10d-f). Klingberg’s protocol may preserve the fluorescence of GFP more than other organic

solvent-based clearing protocols, but it still quenched GFP more severely than our method. Moreover, the signal intensities of endothelial cells stained using anti-CD31 were low, making observations of the microvascular structure in the brain white matter almost impossible, which is not the case with our SeeNet protocol (Supplementary Figure 10g-k, Figure 5b-d). Therefore, we would like the advancement of our method to be re-evaluated.

Comment #2-2

In addition, the authors mention two references which should show that SDC was supposed not to be suitable for clearing in animal tissue (page 6, row 5-6), but these manuscripts do not mention SDC at all.

Answer #2-2

These manuscripts indeed mentioned SDC. In Yang et al. (2014), please see Figure S1 (<https://www.sciencedirect.com/science/article/pii/S0092867414009313?via%3Dihub>). In Susaki et al. (2014), please see the “chemical screening” section, in which the authors stated, “Deoxycholate was the best candidate for clearing detergents in the first screening, but led to EGFP quenching in the presence of aminoalcohol and urea.” (<https://www.sciencedirect.com/science/article/pii/S0092867414004188?via%3Dihub>). In our study, we did not use aminoalcohol and urea. We found that sodium deoxycholate alone was sufficient to delipidate the whole brain and that the fluorescence of dVenus, GFP, and mCherry was preserved after clearing (Supplementary Figure 7 and Figure 4g-i).

Comment #3.

The authors statement that perfusing brains with gelatin and FITC-albumin would produce costs of 200 US\$/sample cannot be verified. The cost of all materials needed for this method averages out at around 30 US\$.

Answer #3

We calculated the cost as follows. According to Tsai et al. (2009), their protocol uses 20 mL of 1% FITC-albumin solution, which requires 200 mg of FITC-albumin (A9771, Sigma-Aldrich). FITC-albumin costs approximately 1,000 US\$ per gram (<https://www.sigmaaldrich.com/catalog/product/sigma/a9771>); thus, this protocol requires at least 200 US\$ per sample. To the best of our knowledge, Sigma-Aldrich A9771 is currently the cheapest FITC-albumin. This information was provided in the original paper and the previous

manuscript (the Method section, P16 L7-22).

Comment #4.

The manuscript mainly provides data obtained from projections of confocal microscope pictures in 2D immunostainings of cutted sections. This clearly counteracts the purpose of whole-organ preparation for 3D analysis. The data provided from light sheet microscopy is recorded in low magnification and therefore the hypothesis of hippocampal and cortex connecting vessels is demonstrated in poor resolution pictures as well as videos.

Answer #4

We indeed provided most of the data using confocal microscopy; however, this does not necessarily mean that SeeNet-treated samples are not suitable for whole-organ vasculature analysis. We showed that our protocol was applicable for whole-brain imaging in Figure 5. In this revision, we increased the number of whole-brain imaging experiments (Supplementary Figure 7).

We also used a macro-zoom light-sheet microscope. As shown in Figure 5b and Supplementary Movie 1, the resolution was sufficiently high to capture the cerebral microvasculature. To more quantitatively discuss this issue, we measured the diameter of capillaries using a confocal microscope (Supplementary Figure 9c). In this imaging condition, one image pixel corresponded to $0.794\ \mu\text{m}$. The mean diameter of the capillary was $6.58 \pm 1.21\ \mu\text{m}$, which was comparable to the spatial resolution of macro-zoom light-sheet microscopy ($6.45\ \mu\text{m}$). These data also suggest that macro-zoom microscopy has sufficient spatial resolution to capture vascular connectivity. Higher-resolution whole-brain imaging is an option when further information is required, for example, for the study of detailed interactions of subcellular structures of glial cells and vasculature at the whole-brain scale. However, especially in whole-brain imaging, choosing the appropriate resolution depending on the purpose of the experiments is important. Too high-resolution imaging makes the data size too large. At the current resolution, the data size for whole-brain imaging with two colors is approximately 12-15 GB per sample, which can be handled in a single workstation. Please note that if the resolution is two times higher, the data size increases 8 times (96-120 GB per sample). We thus think that macro-zoom light-sheet microscopy is one of the best setups to study the connectivity of molecularly characterized cerebral vasculature at the whole-brain level.

Reviewers' comments:

Reviewer #1 (Remarks to the Author):

The authors have successfully answered almost all the issues raised on my previous comment to first version of the manuscript.

Dylight594 instead of Dylight488 (as a positive control for the vasculature staining) could be to performed the needed control.

In 199488_2_art_file_3978449_pvv69v.doc, line 22-23 "normal method " regarding the PFA perfusion/fixation should read better with "standard method" instead.

Reviewer #3 (Remarks to the Author):

Comment #1

See below.

We would like to thank all the reviewers for the positive evaluation toward our revised manuscript. Also, we would like to appreciate very helpful and constructive comments raised by the reviewers. Following the reviewers' advice, we have conducted additional experiment and revised the text. Again, we would like to acknowledge the reviewers' contribution in improving our work. We hope that the revised manuscript is now suitable for publication in *Nature Communications*. The revised parts are highlighted in blue in the manuscript.

Each specific comment is labeled with the heading of "Comment #...", and our response to the comment is shown in blue with the heading "Answer #..".

Reviewer #1 (Remarks to the Author):

Comment #1

Dylight594 instead of Dylight488 (as a positive control for the vasculature staining) could be to performed the needed control.

Answer #1

Thank you for suggesting this experiment. We have carried out an additional experiment, using Dylight594-lectin as a positive control for the vasculature staining (New Supplementary Figure 6). Unlike the immunohistochemical staining against CD31 (Supplementary Figure 4), the performances of lectin labelling were similar between the gelatin-FITC-albumin perfusion method and our method, presumably because the labelling was conducted before the perfusion of warmed PFA and the change in the antigenicity (Supplementary Figure 6c). This procedure enabled quantitative comparisons of the staining performances between the two methods, which indicated that our protocol has slightly higher mean staining performance, due to the occasional clogging observed in the gelatin-FITC-albumin method (Supplementary Figure 6a, d). Again, we would like to appreciate this reviewer's suggestion that enabled the quantitative comparison of the staining performances, and we hope that this result has highlighted another pitfall of the previous method.

Comment #2

In 199488_2_art_file_3978449_pvv69v.doc, line 22-23 "normal method " regarding the PFA perfusion/fixation should read better with "standard method" instead.

Answer #2

Thank you for the correction. We have followed the reviewer's suggestion (P5,L20).

Reviewer #3 (Remarks to the Author):

Comment #1

See below.

Answer #1

In the previous letter, we have provided answers for all the comments.

Dear Reviewers,

We would like to thank all of you for positive evaluation towards our manuscript entitled “Visualization and molecular characterization of whole-brain vascular networks with capillary resolution”, manuscript #NCOMMS-19-05466. Following the editorial requests, we have revised our manuscript one last time. All the revised parts in the main text are indicated using ‘track changes’ feature in Microsoft Word.

We hope that the revised manuscript is now suitable for publication in *Nature Communications*.

Sincerely,

Takeyuki Miyawaki, PhD